# Assessment of the Lateral Vibration Serviceability Limit State of Slender Footbridges Including the Postlock-in Behaviour

**Rocío G. Cuevas *** , **Javier F. Jiménez-Alonso** , **Francisco Martínez** and **Iván M. Díaz**

Department of Continuum Mechanics and Theory of Structures, Universidad Politécnica de Madrid, ETS Ingenieros de Caminos, Canales y Puertos, 280404 Madrid, Spain; jf.jimenez@upm.es (J.F.J.-A.); francisco.martinez@upm.es (F.M.); ivan.munoz@upm.es (I.M.D.)

* Correspondence: r.gcuevas@alumnos.upm.es; Tel.: +34-636-46-88-84

**Abstract:** The lateral vibration serviceability of slender footbridges has been the subject of many studies over the last few decades. However, in spite of the large amount of research, a common criterion has not been set yet. Although the human–structure interaction phenomenon is widely accepted as the main cause of the sudden onset of high amplitudes of vibration, the current design recommendations do not include an expression for the auto-induced component of the pedestrian action and, as a consequence, it is not possible to evaluate the footbridge comfort once the lock-in effect has developed. Hence, the purpose of this paper is to propose a general formulation, which allows the analysis of the different load scenarios that the footbridge will experience during its overall life cycle. An important advantage over most current design guidelines is that the procedure permits the evaluation of the comfort level of the footbridge, even with crowd densities above the "critical number", and thus takes informed decisions about the possible use of external devices to control the vibration response, depending on the probability of occurrence of the problem. The performance of the proposed method is successfully evaluated through numerical response simulations of two real footbridges, showing a good agreement with the experimental data.

**Keywords:** footbridges; vibration serviceability assessment; lateral vibrations; human–structure interaction

---

## 1. Introduction

The prediction of the dynamic response of pedestrian structures is an issue of increasing importance. During the last decades, the construction of slender footbridges has become a growing trend, leading to numerous problems in the lateral vibration serviceability. The most renowned cases are the Passerelle Solférino (Paris, 1999) (now Passerelle Léopold-Sédar-Senghor) [1] and the London Millennium Bridge (2000) [2,3]. Both experienced excessive lateral vibrations during their opening days. However, the problem is not limited to new and original designs. Prior to these two cases, other footbridges with different designs, such as the T-Bridge (Tokyo, 1993) [4] and the footbridge over the Main at Erlach (Germany, 1972) [5,6], had also experienced excessive lateral accelerations. Additionally, the importance of assessing the lateral vibrations of in-service footbridges is justified by the growing numbers of existing pedestrian structures during the last fifty years as a result of the urban development of city suburbs. This has triggered the need to review and adapt these structures to the new traffic demands in order to prevent undesirable situations.

The phenomenon of experiencing excessive lateral accelerations (generally known as the lock-in effect) occurs in low-damped structures with natural frequencies in the range 0.4–1.3 Hz, when the number of pedestrians on the footbridge is above a certain "critical number". This natural frequency

interval is defined according to the recommendations included in the current design guidelines and the analysis performed by Ingólfsson et al. [7], which concluded that footbridges with natural frequencies within this range have the potential to suffer from excessive pedestrian-induced vibrations. The lock-in behaviour is characterised by a sudden onset of high amplitudes of vibrations, usually associated with resonant loads but that can also be explained by the phenomenon of interaction between pedestrians and the structure. Pedestrians are biomechanical systems [8–10] that generate ground reaction forces while walking, due to the acceleration or deceleration of their centre of mass. The lateral force is relatively small (around 4%–5% of the body weight) [11–13], and is the consequence of the action of keeping the body balance during walking. It can thus be affected by the ground stability [14,15]. As a consequence, the effect of pedestrians in a swaying deck is not limited to imposing external loads. As people perceive the floor vibration, they modify their gait in order to avoid losing balance. People interact with the structure generating the auto-induced force in the natural frequency of the footbridge, with independence of the pedestrian frequency. Hence, the human–structure phase synchronisation is not a necessary condition for the development of the resonant component of the pedestrian load.

Laboratory tests that have been performed by several researchers provide evidence of the auto-induced forces. For example, the response spectrum measured by Dey et al. [16], or the load spectrum measured by Ricciardelli and Pizzimenti [11] or Ingólfsson and Georgakis [17]. In all these cases, the auto-induced harmonic (in the natural frequency of vibration) accompanies the harmonics of the pedestrian load. In addition, full-scale crowd pedestrian tests have been performed to verify the auto-induced effect and to determine the critical number of pedestrians needed to trigger it. Some examples are the tests conducted in structures such as the Pedro e Inês footbridge (Coimbra) [18], the Changi Mezzanine Bridge (Singapore Changi Airport) [19,20] or the Lardal footbridge (Norway) [21]. Figure 1 reproduces the experimental results of the full-scale pedestrian tests performed on the Pedro e Inês footbridge. The data points have been joined to show the acceleration trend. It can be observed that, as the number of pedestrians increases, the lateral dynamic response increases in four stages:

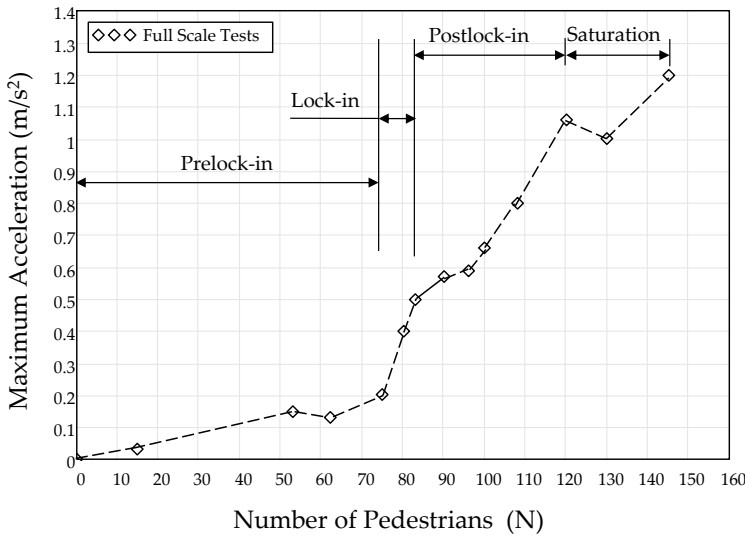

**Figure 1.** Full-scale crowd pedestrian tests on the Pedro e Inês footbridge [18]. Maximum lateral acceleration response vs. number of pedestrians. Stages of lateral vibration response.

1.  Prelock-in: observed in low pedestrian traffic density scenarios. The lateral vibrations are small and slightly perceptible by humans. In this case, the loads exerted on the footbridge are similar to those applied on a rigid surface (not a swaying surface).

2.  Lock-in: it is the instability point, in which the acceleration response builds up suddenly. The number of pedestrians on the footbridge reaches the "critical number" that causes acceleration amplitudes sufficient to trigger the pedestrian–structure interaction phenomenon.

3.  Postlock-in: the crowd density increases above the "critical number". The growth of the structural response accelerates. The auto-induced component of the load must be included when evaluating the footbridge response to pedestrian action.

4.  Saturation: above acceleration levels of around 1–1.2 m/s$^2$ [21], some pedestrians feel uncomfortable and choose to change their gait or stop. The growth of the structural response slows down.

It can therefore be concluded that the human–structure interaction (HSI) phenomenon is widely accepted in the existing literature as the main cause of the lock-in and postlock-in behaviours. However, there are different hypotheses about how to formulate the pedestrian loads and, in consequence, different techniques to evaluate the footbridge response. The pedestrian load is assumed to be a random load with large intra-subject variability (changes in the load exerted by the same person at different times) and inter-subject variability (differences between subjects).

Some recommendations and design guidelines (e.g., Sétra [22], Hivoss [12], Fib [23], Eurocode 5 [24] and Iso-10137 [25]) use a deterministic load model to evaluate the footbridge response. Given that the load takes approximately the same value at equal slots of time periods, and neglecting the intra-subject variability, the single pedestrian action is considered to be a harmonic load, in resonance with the bridge vibration. The effect of a group of pedestrians is taken into account by considering an effective number of perfectly synchronised pedestrians [12,22,23]. The main advantage of the deterministic model is that it is simple to use for design purposes. However, as pointed out by Ingólfsson [7] and Ricciardelli [26], the deterministic methods are not reliable for groups of pedestrians as a result of the great scatter of effective numbers of pedestrians given by the different guidelines. For example, for 100 pedestrians, the effective number of pedestrians would range from 10 to 19 [7,26]. Furthermore, the deterministic methods fail to consider the HSI phenomenon. Even though Hivoss [12] and Sétra [22] introduce the modification of the modal frequency due to the HSI by modelling the pedestrians as passive masses, they fail to account for the influence on modal damping. In addition, current design recommendations lack an expression for the auto-induced component of the pedestrian action and, in consequence, it is not possible to evaluate the footbridge degree of comfort once the lock-in effect has developed. In addition, the codes also have to introduce a criterion for checking the lateral lock-in. Arup´s stability criterion [3] is the most commonly accepted, but it fails to agree with some experimental results. There are other criteria, such as Newland´s [27], Strogatz´s [28] and Roberts´ [29]. Ricciardelli [26] has compared the different formulations, finding again a large scatter in the results. A different approach is to define the threshold acceleration amplitude beyond which the phenomenon develops. Most of the design guides set it between 0.1 and 0.15 m/s$^2$ [12,22,30].

A different methodology, which is formulated in the frequency domain, is to define the pedestrian load as a stationary random process through its power spectral density (PSD), treating the pedestrian flow probabilistically and thus considering the intra- and inter-subject variability of the load. The stationary response is obtained by calculating the root mean square (RMS) value of the modal acceleration. Ricciardelli and Pizzimenti [11] and Ingólfsson et al. [31] have proposed a characteristic PSD for the first five harmonics of the pedestrian lateral action that has been obtained measuring the lateral pedestrian forces on an instrumented treadmill. HSI is included through an experimental coefficient, usually being noted as $c_p$ for the auto-induced in-phase component of the pedestrian force and $q_p$ for the out-of-phase component [31,32], thus considering the change of the modal parameters of the structure in a simplified way. Nevertheless, the spectral methods apply to a stationary and uniform pedestrian crowd sufficiently low to omit the human–human interaction. Moreover, the Hivoss guideline [12] incorporates a response spectra method based on Monte Carlo simulations, but it assumes that the mean step frequency of the pedestrian stream is in resonance with the footbridge dominant mode.

Finally, it is worth mentioning the more elaborate models that also introduce the human–human interaction as part of the HSI phenomenon. For example, Carroll et al. [33] propose a microscopic model of the crowd, with each individual modelled as an "inverted pendulum", inspired by the work

of Barker [34], Macdonald [14] and Bocian [35]. Then, the individual movement as a member of the crowd is modelled using the concept of "social forces" [36], which determine the pedestrian position. A similar approach is presented by Jimenez-Alonso et al. [37], but simulating the pedestrian as a spring-dashpot-mass model. The main advantage of these models is that they accurately reproduce the HSI phenomenon, as they consider most of the multiple factors that intervene. Nevertheless, they are complex and cannot be easily applied yet. In addition, as the authors have noted [14,31], the inverted pendulum equations are very sensitive to the balance law selected, and there is no general agreement about the value of the parameters of the spring-dashpot-mass model that represent the pedestrians [38].

Therefore, it can be concluded that, in spite of the research of the last decades, common criteria for the lateral assessment of footbridges have not been accepted yet. While deterministic models are the more popular ones, they show a great scatter of the effective number of pedestrians given by different guides and consider the HSI phenomenon only partially. Similarly, spectral methods represent better the pedestrian load and the HSI phenomenon, but they are barely represented in the design guidelines. Finally, the more elaborate models are complex and not available in practical engineering applications yet.

A footbridge experiences different load scenarios in its overall life cycle. The comfort level, or structural acceleration response, that users are willing to accept for a specific traffic scenario depends on the probability of its occurrence. Controlling the vibration response usually involves increasing the structural damping by external devices, which has an additional cost. The main purpose of this paper is to propose a simple but general formulation, useful for practical engineering applications, to evaluate the lateral vibration serviceability limit state of slender footbridges, even when the crowd density exceeds the "critical number" (postlock-in behaviour). This tool allows informed decisions to be made about the use of external devices to control the vibration response, or whether not to do anything if such crowds will be encountered only rarely.

The key finding of the study is that the proposed method, based on the frequency domain analysis, allows the response due to the auto-induced component of the pedestrian load to be expressed in terms of an amplification ratio of the response with no interaction. The ratio only depends on the dynamic properties of the structure, making it very simple to estimate the dynamic response and the structural damping required for any possible pedestrian traffic scenario, as well as to analyse the footbridge sensitivity to the HSI phenomenon.

The paper continues with the description of the proposed formulation in Section 2. Then, the practical application of the method is presented in Section 3. Finally, the main conclusions and some future works are given in Section 4.

## 2. Method to Evaluate the Lateral Vibration of Footbridges Including the Postlock-in Behaviour

The prediction of the lateral response of footbridges is a complex issue, given that it is difficult to set an analytical expression for the pedestrian action that considers the multiple factors involved. The challenge is to find a procedure sufficiently accurate for evaluating the response and easy to apply in practical engineering applications. Therefore, the evaluation of the lateral dynamic response is performed in the frequency domain, a method especially efficient for representing the pedestrian action as a harmonic load. Furthermore, frequency domain analysis allows superposition, which simplifies the formulation.

Another advantage is that the structural response can be evaluated independently for the particular mode of vibration in which the lateral instability is more likely to develop, omitting the possible interaction between modes. The response is calculated using an equivalent single degree of freedom (SDOF) system, where the structure is characterised through the system's complex frequency response

function $H(f)$ (mN$^{-1}$), which defines completely the dynamic characteristics of the footbridge for the mode considered, and takes the well-known expression:

$$H(f) = \frac{1}{K - M(2\pi f)^2 + iC(2\pi f)} \tag{1}$$

where $f$ is the variable frequency (Hz), $K$ the modal stiffness (Nm$^{-1}$), $M$ the modal mass (kg) and $C$ the modal damping (Nsm$^{-1}$). For the particular case of lightly damped footbridges, $H(f)$ presents a sharp peak at $f = f_b$ (the footbridge's natural frequency) and tends to zero when moving away from this value. Hence, the structural response is amplified only in a very narrow frequency interval centred on $f_b$.

The pedestrian flow is modelled by its average properties: the number of pedestrians $N$ and the pedestrian´s lateral step frequency $f_p$ (Hz). The step frequency $f_p$, is a random variable with a normal distribution of probability $P(f_p)$ (mean $\mu = 0.86$ Hz and standard deviation $\sigma = 0.08$ Hz [39,40]). Other factors that also influence the pedestrian action (such as the walking speed or the step length) are considered to be represented by the step frequency and the geometric relations between them [39–41], and thus are not taken into account for characterising the pedestrian flow. In this paper, the step frequency $f_p$ is taken in the interval $(\mu-3\sigma, \mu+3\sigma)$, which represents a confidence level of 99.7%, which is adequate for the accuracy of this study, since it nearly covers the complete pedestrian´s excitation range without compromising the computational cost of the problem.

The HSI phenomenon (considered to be the main cause of the onset of high amplitudes of vibrations) is explained using a linear feedback model similar to that proposed by Newland [3]. Figure 2 illustrates the feedback system. In Figure 2, the interval of the footbridge natural frequency is defined according to references [7,12,22]. With low pedestrian traffic density (prelock-in stage), the modal displacement response $Y(f)$ is obtained as the algebraic product of the modal pedestrian force $X(f)$ exerted by the pedestrians with no bridge motion and the frequency response function $H(f)$. However, as the number of pedestrians grows above the "critical number" (postlock-in stage), the footbridge movement affects the pedestrians´ gaits. The function $\alpha(f)$ is the feedback function by which the modal input force $X(f)$ is amplified by the footbridge response $Y(f)$. The modal displacement response $Y(f)$ is obtained in this stage as the algebraic product of the total modal pedestrian force $X(f)+\alpha(f)xY(f)$ and $H(f)$.

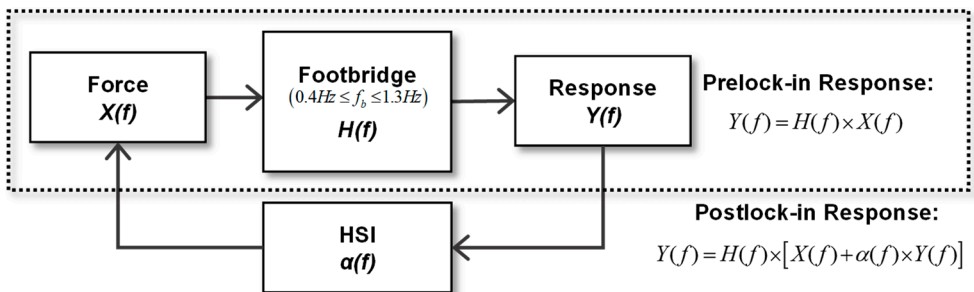

**Figure 2.** Feedback system to model human–structure interaction (HSI).

The functions $X(f)$ and $\alpha(f)$, which define the pedestrian action, are adopted from the results of the extensive experimental campaigns that were performed by Ingólfsson et al. [17,31], given that in these campaigns both pedestrians´ actions on static and moving platforms were measured. Therefore, it is possible to define both functions using the same source and, in addition, the experimental results are introduced in the numerical formulation. However, given that the measurements were taken in a laboratory, the method does not take into account the human–human interaction effect that occurs with high traffic densities. In the next subsections, the numerical values of $X(f)$ and $\alpha(f)$ are obtained and the footbridge response is calculated.

*2.1. Footbridge Response Due to the Auto-Induced Component of the Pedestrian Load*

As described before, when walking on a moving surface, people perceive the floor vibration and modify their gait to maintain balance. This interaction with the motion of the structure produces the auto-induced components of the pedestrian load. In the case of lightly damped footbridges, the structural response is amplified only in a very narrow frequency interval, and the resonant harmonic of the auto-induced force induces the maximum dynamic response in the footbridge's natural frequency. Experiments show that this resonant harmonic is independent of the pedestrian frequency. Even if its magnitude is small for one individual, the passage of a crowd can trigger a multiplying effect. Hence, the proposed method assumes that the auto-induced response is controlled by the resonant harmonic, omitting other components such as that proportional to acceleration, which slightly modifies the natural frequency but affects little the response magnitude.

In the frequency domain, the spectral value of the auto-induced response can be directly obtained once the spectral value of the auto-induced resonant harmonic is known. Considering that the steady-state velocity response of a damped system due to a resonant harmonic vibration ($f = fb$) is in-phase with the load, the resonant force can be expressed through the velocity proportional component of the experimentally measured actions. Hence, the auto-induced component $\alpha(f) \times Y(f)$ in Figure 2 is defined from the dynamic tests performed by Ingólfsson et al. in 2009 at the University of Florence in Prato, Italy [31]. Seventy-one volunteers participated by walking on a swaying treadmill with lateral harmonic displacement expressed by $y(t) = y_0 \, sin(2\pi f_b t)$ (m), where $y_0$ (m) is the amplitude of the lateral movement and $f_b$ (Hz) is the frequency. Ingólfsson et al. measured both the treadmill motion and the lateral forces that were exerted by the participants. The auto-induced components, which were determined through the cross-covariance between the measured pedestrian force and the velocity of the treadmill, were expressed as proportional to the treadmill velocity in the following way:

$$c_p\left(\frac{f_b}{f_p}, y_0\right) y'(t) \quad \overset{Fourier\ Transform}{\underset{\longleftarrow}{\longrightarrow}} \quad c_p\left(\frac{f_b}{f_p}, y_0\right) 2\pi f \, Y(f) \tag{2}$$

in which the coefficient of proportionality $c_p(f_b/f_p, y_0)$ (Nsm$^{-1}$) that depends on the frequency ratio $f_b/f_p$ and on the amplitude of the movement $y_0$ is defined by fitting an exponential function to the data [17], and thus considering the large randomness of the measured load. Therefore, the feedback function $\alpha(f)$ (Nm$^{-1}$) can be obtained from Equation (2) and expressed as follows:

$$\alpha(f) = c_p\left(\frac{f_b}{f_p}, y_0\right) 2\pi f \tag{3}$$

In order to present an easy formulation, the coefficient of proportionality is defined by using the simplest expression proposed by Ingólfsson [42]. This expression omits the dependency with the amplitude $y_0$ and is expressed as follows:

$$c_p\left(\frac{f_b}{f_p}\right) = -794\left(\frac{f_b}{f_p}\right)^2 + 1558\frac{f_b}{f_p} - 580 \, ; \quad 0.4 \le \frac{f_b}{f_p} \le 1.2 \tag{4}$$

Figure 3 compares the value of the coefficient of proportionality $c_p$ obtained using the amplitude dependent stochastic model [17] at four different vibration amplitudes (4.5, 10, 19.4 and 31 mm), with the value obtained using Equation (4) [42]. It can be observed that the value of $c_p$ decreases as the amplitude increases, resulting in the second order polynomial (Equation (4)) being an upper bound of the experimental results.

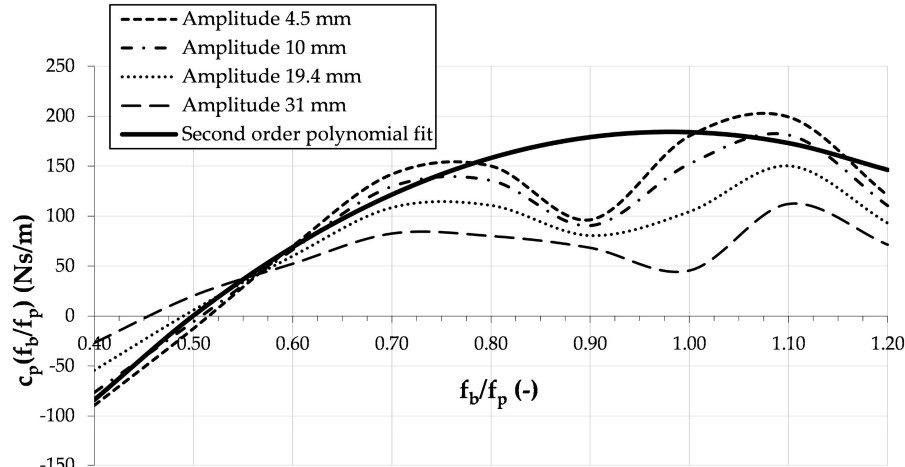

**Figure 3.** Statistical characterisation of damping coefficient *cp(fb/fp)* lateral amplitude dependent [17] vs. polynomial best fit lateral amplitude independent [42].

Furthermore, the dependence of the coefficient $c_p(f_b/f_p)$ on the step frequency $f_p$ shows the inter-subject variability in the auto-induced loads. In a real case assessment, there is a crowd of pedestrians walking randomly with different frequencies $f_p$. Hence, using the probability distribution function of the step frequency $P(f_p)$, and applying the superposition principle, the contribution of each possible value of $f_p$ included in the interval $(\mu-3\sigma, \mu+3\sigma)$ in the auto-induced load exerted by one pedestrian is expressed through the coefficient $c_p(f_b)$ (Nsm$^{-1}$), as follows:

$$c_p(f_b) = \int_{\mu-3\sigma}^{\mu+3\sigma} c_p\left(\frac{f_p}{f_b}\right)P(f_p)df_p; \quad 0.4 \leq f_b \leq 1.3 \tag{5}$$

Figure 4 shows the value of the coefficient $c_p(f_b)$ calculated through Equation (5). It can be seen that between 0.42 and 1.23 Hz, the resonant harmonic of the auto-induced pedestrian force is positive, limiting the footbridge natural frequency interval of possible instability.

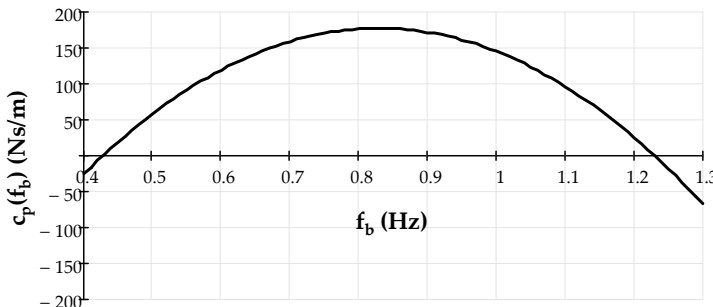

**Figure 4.** The damping coefficient *cp(fb)* independent of the pedestrian step frequency $f_p$ obtained through Equation (5).

The next step is to calculate the spectral value of the auto-induced resonant harmonic exerted by one single pedestrian $|F(f_b)|$ expressed in Newton (N), which can be computed by the product of the coefficient $c_p(f_b)$ (Equation (5)) and the amplitude of the footbridge velocity ($y_0' = 2\pi f_b y_0$ (m/s)), as follows:

$$\left|F(f_b)\right| = c_p(f_b)2\pi f_b y_0 \tag{6}$$

In the real case of *N* pedestrians uniformly distributed on a span of length *L* (m), assuming that the studied vibration mode is the first one (mode shape sine lateral wave of effective length $L_d$ (m)),

the equivalent spectral value of the auto-induced resonant harmonic exerted by $N$ pedestrians $|F_N(f_b)|$ in Newton (N) is expressed in modal coordinates, as follows:

$$\left|F_N(f_b)\right| = \frac{N}{L}\left(\int_0^{L_d} \sin\frac{\pi x}{L_d}dx\right)c_p(f_b)2\pi f_b u_{0,N} = N\frac{L_d}{L}4f_b c_p(f_b)u_{0,N} \tag{7}$$

in which $u_{0,N}$ is the modal displacement amplitude of the footbridge due to $N$ pedestrians before the HSI develops. Consequently, the spectral value of the auto-induced displacement response, $|Y_N(f_b)|$, is computed by the product of the spectral value of the auto-induced resonant harmonic exerted by $N$ pedestrians, $|F_N(f_b)|$ (Equation (7)), and the resonant amplitude of the complex response function, $|H(f_b)|$ (Equation (1) particularised in the natural frequency $f_b$), resulting:

$$\left|Y_N(f_b)\right| = \left|F_N(f_b)\right|\left|H(f_b)\right| = N\frac{L_d}{L}4f_b c_p(f_b)u_{0,N}\left|H(f_b)\right| \tag{8}$$

Considering Equation (8), the amplitude of the resonant displacement response in the time domain can be computed doubling the spectral value $|Y_N(f_b)|$. Consequently, the amplitude of the resonant acceleration response due to the auto-induced component $a_{v,N}$ (m/s$^2$) can be expressed in terms of amplification ratio to the modal acceleration amplitude $a_{0,N}$ (m/s$^2$) without pedestrian–structure interaction, as follows:

$$\frac{a_{v,N}}{a_{0,N}} = N\left[\frac{L_d}{L}8f_b c_p(f_b)\left|H(f_b)\right|\right] \tag{9}$$

The right side of Equation (9) has two factors. One is the number of pedestrians and the other depends on the modal properties of the structure, which are constant for a specific mode of vibration. Thus, Equation (9) can be rewritten, showing a positive relation between the amplification ratio $a_{v,N}/a_{0,N}$ and the number of pedestrians $N$:

$$\frac{a_{v,N}}{a_{0,N}} = N\,G \tag{10}$$

in which, establishing a particular value for the variables $L_d$, $L$, $f_b$, $c_p(f_b)$ and $|H(f_b)|$, the non-dimensional constant $G$ (-) is computed as follows:

$$G = \frac{L_d}{L}8f_b c_p(f_b)\left|H(f_b)\right| \tag{11}$$

Equation (10) shows that the auto-induced response depends on the initial level of perceived vibration-triggering HSI. The following subsection deals with the determination of this initial value.

## 2.2. Footbridge Response without Bridge Motion

With low traffic density or traffic density that is below the "critical number" of pedestrians, the footbridge response does not affect people balance and does not induce HSI. The loads exerted on the footbridge are similar to those measured on a static treadmill. Hence, the function $X(f)$ (N) in Figure 2 is expressed through the experimental power spectral density (PSD) of the load that was obtained by Ingólfsson and Georgakis [17] in the absence of lateral motion, thus considering the intra-subject variability of the load in the analytical expression. The PSD is based on a Gaussian-shaped function to fit the individual load harmonics, which were previously proposed by Pizzimenti and Ricciardelli [11]. Equation (12) is the expression of the mean or the 95% fractile PSD function $S(f,f_p)$ (N$^2$Hz$^{-1}$) for the first five harmonics [42]. In this paper, the lateral step frequency $f_p$ is introduced as a variable, which means that Equation (12) represents the PSD of the load exerted by any pedestrian who walks with any frequency $f_p$:

$$S(f,f_p) = \sum_{j=1}^{5}\left(\frac{2A_j\sigma_j^2 W^2}{\sqrt{2\pi}B_j f}e^{-2\left(\frac{\frac{f}{jf_p}-1}{B_j}\right)^2}\right) \tag{12}$$

The parameters $A_j$, $B_j$ and $\sigma_j$ are obtained from the experimental data fit. $W = 700$ N is the mean of people weight. The variance $\sigma_j{}^2$, which represents the energy content in the load around the j-harmonic, can be taken as either the data mean value, in order to represent the mean of the pedestrian load, or the 95% fractile (or that with 5% probability of exceedance), in order to represent the peak value of the pedestrian load. The values $A_j$, $B_j$ and $\sigma_j$ are summarised in Table 1.

**Table 1.** Parameters for the Gaussian-shape spectrum $SF(f,f_p)$ [42].

|  | j = 1 | j = 2 | j = 3 | j = 4 | j = 5 |
|---|---|---|---|---|---|
| $A_j$ | 0.900 | 0.020 | 0.774 | 0.0258 | 0.612 |
| $B_j$ | 0.043 | 0.031 | 0.026 | 0.064 | 0.026 |
| $\sigma_j/W$ (mean value) | 0.035 | 0.005 | 0.018 | 0.004 | 0.008 |
| $\sigma_j/W$ (95% fractile) | 0.054 | 0.008 | 0.025 | 0.006 | 0.0012 |

Furthermore, the dependence of the function $S(f,f_p)$ with the step frequency $f_p$ shows the inter-subject variability in the loads. As done before, using the probability distribution function of the step frequency $P(f_p)$ and applying the superposition principle, the contribution of each possible value of $f_p$ included in the interval $(\mu-3\sigma, \mu+3\sigma)$ of the load exerted by one pedestrian is expressed through the function $SF(f)$ (N$^2$), which can be expressed independently from the step frequency, as follows:

$$SF(f) = \int_{\mu-3\sigma}^{\mu+3\sigma} S(f,f_p)P(f_p)df_p \tag{13}$$

Figure 5 shows the first and third harmonics determined using Equation (13), centred in $1f_p = 0.86$ Hz (mean value of the step frequency) and $3f_p = 2.58$ Hz. The second, fourth and fifth harmonics are nearly zero as the even harmonics are small because of the asymmetric character of the pedestrian step (Table 1) and $P(f_p)$ tends to zero outside of the interval 0.62 Hz and 1.10 Hz.

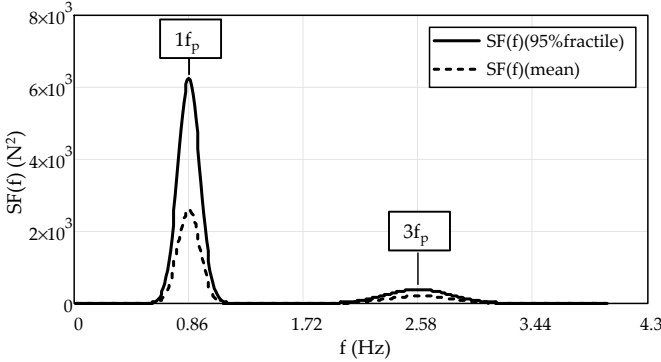

**Figure 5.** Experimental mean and 95% fractile power spectral density functions $SF(f)$ of the pedestrian load exerted on static surface expressed independently of the pedestrian frequency.

For the case of one pedestrian that is uniformly distributed on the footbridge span, the equivalent PSD of the load $SX(f)$ (N$^2$) is expressed in modal coordinates as follows:

$$SX(f) = \left(\frac{1}{L}\int_0^{L_d} \sin\frac{\pi x}{L_d}dx\right)^2 SF(f) = \left(\frac{1}{L}\frac{2L_d}{\pi}\right)^2 SF(f) \tag{14}$$

The next step is to determine the mean or peak value of the PSD of the lateral response $SY(f)$ (m$^2$) by using the following expression:

$$SY(f) = |H(f)|^2 SX(f) \tag{15}$$

Equation (15) is represented in Figure 6 by overlapping the functions $SX(f)$ and $|H(f)|^2$ in the case of the Pedro e Inês footbridge [18] with $f_b$ = 0.9 Hz, $M$ = 165,880 kg, $K$ = 5,304,000 Nm$^{-1}$, $C$ = 10,880 Nsm$^{-1}$, $L$ = 144 m and $L_d$ = 88 m. It can be observed that the structural response is amplified only in a very narrow interval centred in $f_b$. Additionally, in most of the cases (footbridges with a natural frequency in the range 0.4–1.3 Hz), the dynamic response is only caused by the PSD of the first harmonic of the load.

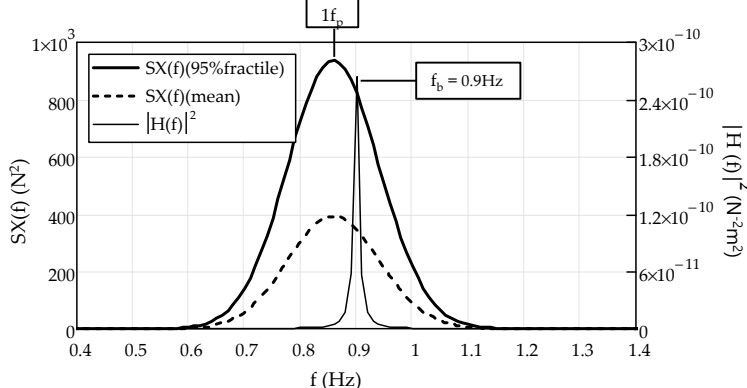

**Figure 6.** The functions $SX(f)$ ($L$ = 144 m and $Ld$ = 87.5 m) and $H(f)^2$ ($f_b$ = 0.9 Hz, $M$ = 165,880 kg, $K$ = 5,304,000 Nm$^{-1}$ and $C$ = 10,880 Nsm$^{-1}$ [18]) are shown overlapped to represent Equation (15).

Consequently, the mean $a_{0,mean}$ (m/s$^2$) and maximum $a_{0,max}$ (m/s$^2$) amplitude of the acceleration response without bridge motion due to one pedestrian uniformly distributed can be expressed by calculating the root mean square (RMS) value of the PSD response $SY(f)$, as follows:

$$a_{0,mean} = 4\pi^2 f_b^2 \sqrt{2 \int_0^{f_{Ny}} (SY(f))_{mean} df} \tag{16}$$

$$a_{0,\max} = 4\pi^2 f_b^2 \sqrt{2 \int_0^{f_{Ny}} (SY(f))_{\max} df} \tag{17}$$

where $f_{Ny}$ (Hz) is the Nyquist frequency. Thus, the probabilistic response is integrated over the frequency range to determine a single expected mean or maximum value. Equation (17) is used to calculate the maximum value of the acceleration response without bridge motion, while Equation (16) is used to calculate the mean value of the acceleration response considered to be perceived by pedestrians, in order to compare it with the threshold beyond which the HSI develops.

### 2.3. Footbridge Response: General Formulation

As mentioned in Section 1, the purpose of this paper is to propose an easy-to-apply but general formulation that permits the footbridge lateral response in the four stages of the response as the number of pedestrians grows to be estimated (Figure 1). The evaluation of the dynamic response is performed in the frequency domain, and thus the steady state response of the structure to a certain steady state input is determined. This consideration is equivalent to assuming that all pedestrians remain in a stationary position with respect to the footbridge, exerting an oscillatory load in their respective locations. It is considered that most pedestrians will perceive the mean acceleration response. If this is sufficient to affect people's gaits, HSI develops and the pedestrian load increases due to the auto-induced components. Another number of pedestrians is treated independently, as a different load case.

Therefore, for each of the four stages described in Section 1, the proposed formulation is described in Figure 7 and it may be summarised in the following steps:

1.  Prelock-in: the maximum acceleration response is calculated by using Equation (17).
2.  Lock-in: the criterion for the lateral lock-in is suggested to establish that the acceleration threshold beyond which HSI develops is 0.1–0.15 m/s$^2$. When pedestrians perceive this acceleration level, they modify their gait and interact with the structure. The critical number of pedestrians is determined by comparing the mean value of the acceleration response calculated by using Equation (16), with 0.125 m/s$^2$.
3.  Postlock-in: the maximum acceleration response is calculated by adding the response without bridge motion (Equation (17)) and that due to the auto-induced component of the load (Equation (10)). In Equation (10), the initial level of vibration that people perceive $a_{0,N}$ is obtained using Equation (16).
4.  Saturation: the acceleration response is limited to the value 1.2 m/s$^2$ [7,21].

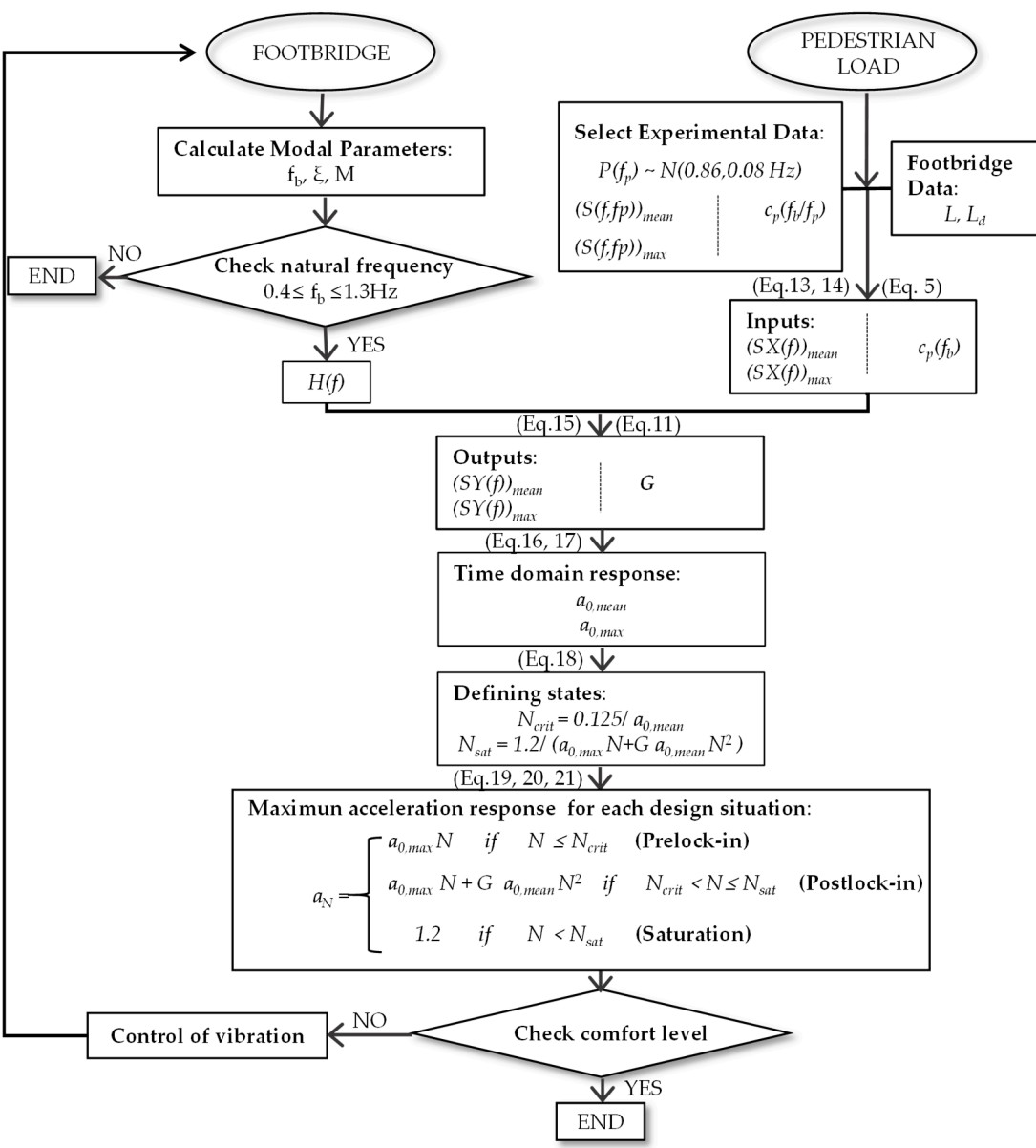

**Figure 7.** Flowchart for the application of the proposed formulation.

Consequently, the critical number of pedestrians $N_{crit}$ and the maximum acceleration response $a_N$ (m/s$^2$) is expressed as follows:

$$N_{crit} = \frac{0.125}{a_{0,mean}} \tag{18}$$

$$a_N = a_{0,max}N \quad if \ \ N \le N_{crit} \tag{19}$$

$$a_N = a_{0,max}N \ + \ G\,a_{0,mean}\,N^2 \ \ if \ \ N_{crit} < N \le N_{sat} \tag{20}$$

$$a_N = 1.2 \ \ if \ \ N_{sat} < N \tag{21}$$

## 3. Numerical Applications

The performance of the model is evaluated through the numerical response simulations of two real footbridges: the Pedro e Inês footbridge [18] and the Lardal footbridge [21]. Full-scale crowd pedestrian tests were performed in both, making it possible to compare the predictions of the proposed method with the results of the tests. Additionally, the lateral response of the two structures is calculated using the Hivoss guideline [12] and also compared with the proposed method.

### 3.1. Pedro e Inês Footbridge

The Pedro e Inês footbridge (Figure 8) is located in Coimbra (Portugal), above the river Mondego. It was opened in November 2005. The 274.5 m long footbridge is formed by a central parabolic arch of 9.4 m rise and 110 m chord, two lateral semi-arches of 64 m and shorter approach spans at each end. The bridge deck is made of a steel concrete composite box girder of 4 m width and the arches are made of steel box girders. The singularity of this footbridge is the anti-symmetry of both arches and deck cross-sections along the longitudinal axis of the bridge [18]. Preliminary calculations indicated that the footbridge was prone to vibrations induced by pedestrians, both in lateral and in vertical directions.

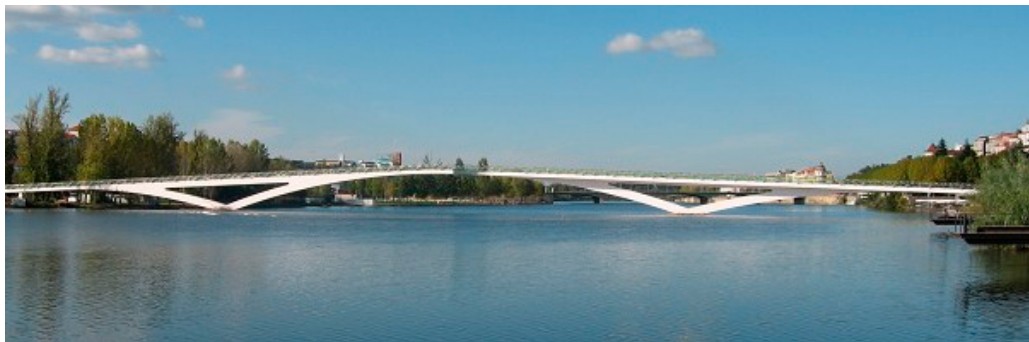

**Figure 8.** The Pedro e Inês footbridge. (Picture after https://structurae.net/en/structures/pedro-and-ines-bridge.).

In April 2006, ambient vibration tests were performed, with 20 sections of the bridge instrumented to identify the natural modes of vibration [43]. The frequency of the first lateral mode was determined to be 0.91 Hz, with a damping ratio between 0.5%–0.6% and a modal mass in the state of completion by the time when the tests were performed, around 165,880 kg. The mode shape resulted to be a lateral sine wave. The wavelength was estimated from the experimental mode shape, considering the location of the instrumented sections (between section 7 and section 13 [18]), resulting to be around 88 m.

The confirmation of a lateral mode in the critical range motivated the performance of tests with pedestrians. The pedestrians walked freely on the bridge between the instrumented sections 5 and 15 that represent a length of 144 m, with an increasing and controlled number up to a maximum of 145 pedestrians. Figure 10 shows the maximum lateral acceleration recorded at the mid-span as the number of pedestrians increased [18], with a maximum of 1.2 m/s$^2$ that occurred with 145 people.

### 3.2. Lardal Footbridge

The Lardal footbridge (Figure 9) is located in Lardal (Norway), above the river Lagen. On the opening day in 2001, the phenomenon of lateral vibrations was observed at this bridge [21]. The bridge has a shallow arch of 91 m chord, made of glue-laminated timber with steel cable reinforcements in certain parts of the main span.

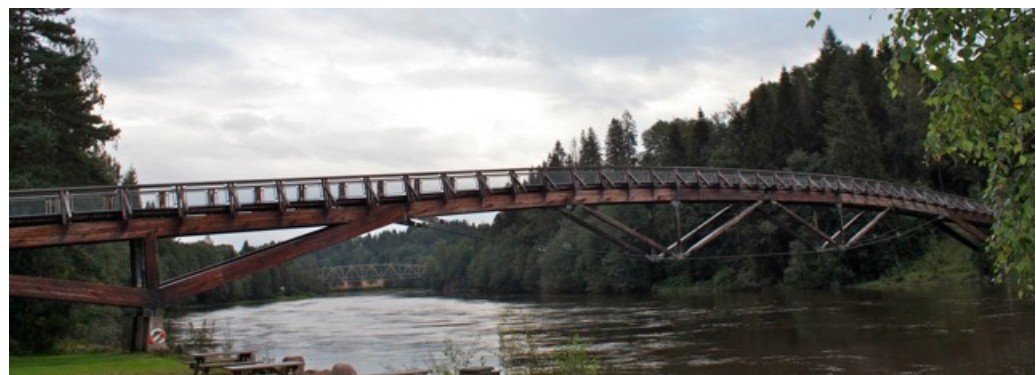

**Figure 9.** The Lardal footbridge. (Picture after http://broer-vestfold.blogspot.com/2011/08/bro-v-kjrra-fossepark-i-lardal.html.).

The modal properties of the first lateral mode were obtained through ambient tests. The natural frequency was determined to be 0.83 Hz, with a damping ratio of 2.5% and a modal mass of around 18,000 kg. The mode shape resulted to be a lateral sine wave with wavelength of 80 m.

Full-scale pedestrian tests were performed to compare the predictions with the measurements. The maximum lateral acceleration recorded at the structure mid-span versus the number of pedestrians is represented in Figure 11. The footbridge resulted to be extremely lively, with an acceleration response exceeding 1 m/s$^2$ for as few as 40 people. During excessive lateral vibrations (beyond 1 m/s$^2$), it was observed that pedestrians accepted accelerations up to a given limit, beyond which some of them decided to stop, thereby limiting the accelerations (saturation level) [21].

### 3.3. Numerical Response Evaluation

Following, the numerical assessment of the maximum acceleration response at the mid-span was performed by using the method proposed in this paper. The modal properties of both structures, which constitute the method data, are summarised in Table 2, where $\xi$ (-) is the damping ratio. The parameters of Equations (18–20) that establish the analytical relation between the maximum acceleration response and the number of pedestrians were calculated by using Equations (1,5,11,16,17). These parameters are summarised in Table 3. The saturation level or acceleration value, beyond which some pedestrians feel uncomfortable and choose to stop, was assumed to be 1.2 m/s$^2$, consistent with the recommendations of some authors [7,21] and the values recorded in the experiments on both footbridges. Equations (18–21) are represented in Figure 10 (the Pedro e Inês footbridge) and Figure 11 (the Lardal footbridge).

**Table 2.** Numerical response simulation. Data [18,21].

| Footbridge | $f_b$ (Hz) | $M$ (kg) | $\xi$ (-) | $L$ (m) | $L_d$ (m) | $b$ (m) |
|---|---|---|---|---|---|---|
| Pedro e Inês | 0.91 | 165,880 | $5.80 \times 10^{-3}$ | 144.00 | 88.00 | 4.00 |
| Lardal | 0.83 | 18,000 | $2.50 \times 10^{-2}$ | 91.00 | 80.00 | 2.40 |

**Table 3.** Numerical response simulation. Results obtained by applying the proposed method.

| Footbridge | $\|H(f_b)\|$ $(mN^{-1})$ | $c_p(f_b)$ $(Nsm^{-1})$ | $G$ (-) | $a_{0,mean}$ $(m/s^2)$ | $a_{0,max}$ $(m/s^2)$ | $N_{crit}$ (-) | $N_{sat}$ (-) |
|---|---|---|---|---|---|---|---|
| Pedro e Inês | $1.59 \times 10^{-5}$ | 170.09 | $3.20 \times 10^{-2}$ | $1.64 \times 10^{-3}$ | $2.53 \times 10^{-3}$ | 75 | 129 |
| Lardal | $4.09 \times 10^{-5}$ | 177.36 | $5.5 \times 10^{-2}$ | $9.52 \times 10^{-3}$ | $15.00 \times 10^{-3}$ | 13 | 36 |

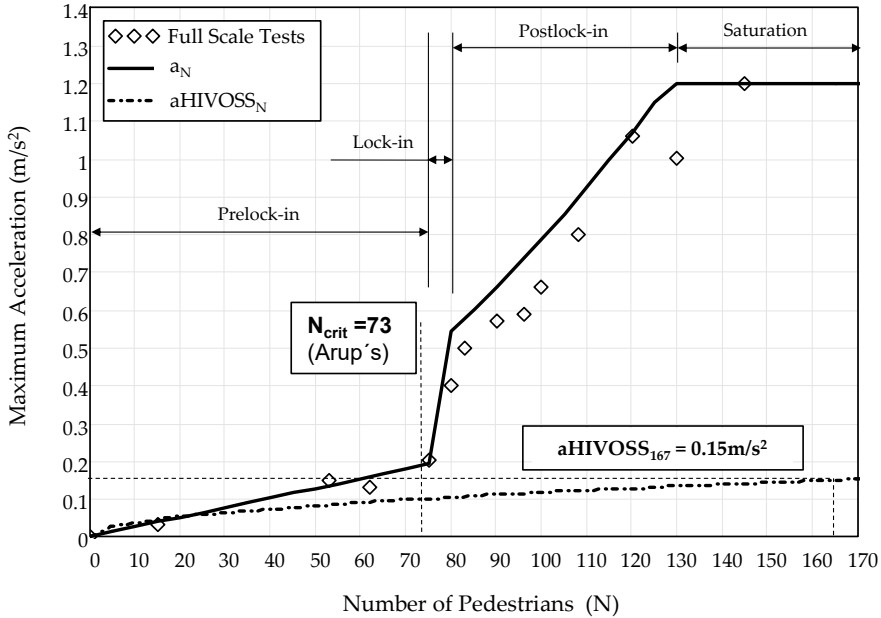

**Figure 10.** The Pedro e Inês footbridge. Analysis of the results.

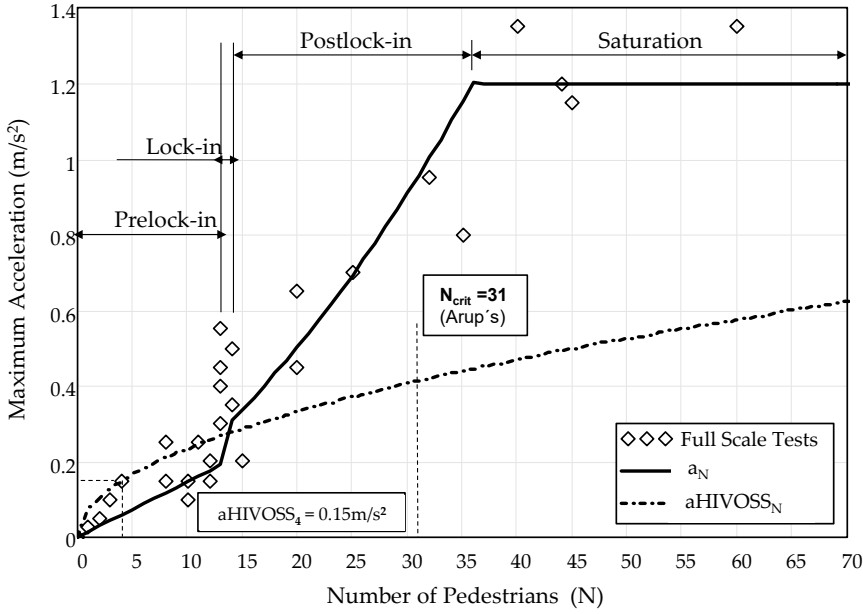

**Figure 11.** The Lardal footbridge. Analysis of the results.

Additionally, the numerical evaluation of the lateral response of the two structures was calculated following the Hivoss guideline [12]. The assessment was also performed in the frequency domain. The structural response was evaluated by using an equivalent SDOF system, in which the structure was characterised through its modal properties (Table 2). The stochastic stream of $N$ pedestrians was defined through the idealised stream of $N'$ perfectly synchronised pedestrians modelled as a harmonic deterministic load. As a result, the analysis provided the maximum resonant acceleration

response at the centre of the footbridge. Table 4 summarises the values of the load $F_0$* (N) due to an idealised stream of $N'$ perfectly synchronised pedestrians, expressed in generalised coordinates; the parameter $a_{max}$ (m/s$^2$), which controls the relation between the acceleration response and the number of pedestrians, also represented in Figures 10 and 11; and the critical number of pedestrians obtained by using the two criteria of the Hivoss guideline. The relation between the acceleration response and the number of pedestrians is:

$$aHIVOSS_N = a_{\mathrm{max}} \sqrt{N} \; if \; d \le 1p/m^2 \tag{22}$$

**Table 4.** Numerical response simulation. Results obtained by applying Hivoss guideline.

| Footbridge | $|H(f_b)|$ (mN$^{-1}$) | $F_0$* (N) | $a_{ma}$ (m/s$^2$) | $N_{crit}$ (Arup´s) (-) | $N_{crit}$ (acc.) (-) |
|---|---|---|---|---|---|
| Pedro e Inês | $1.59 \times 10^{-5}$ | 11.20 | $1.20 \times 10^{-2}$ | 73 | 167 |
| Lardal | $4.09 \times 10^{-5}$ | 33.45 | $7.40 \times 10^{-2}$ | 31 | 4 |

Since in both footbridges a pedestrian density greater than 1 p/m$^2$ implies a high number of pedestrians that is beyond the limits of the full-scale crowd pedestrian tests (576 pedestrians in the case of Pedro e Inês and 218 pedestrians in Lardal), only the expression for pedestrian density lower than 1 p/m$^2$ was considered in this paper.

### 3.4. Analysis of the Results

As exposed previously, Figure 10 shows the results of the full-scale crowd pedestrian tests performed on the Pedro e Inês footbridge, together with the numerical values obtained by applying the proposed method and by following the Hivoss guideline [12]. It can be observed that in the prelock-in stage, a good agreement was achieved between the experimental data and the proposed method that, in addition, predicted very well the lock-in point (75 pedestrians). Furthermore, the proposed method permits the response after HSI develops to be estimated. In the postlock-in stage the analytical line follows and constitutes an upper bound of the experimental data.

Although the Hivoss guideline represented adequately the prelock-in stage, particularly below 30 pedestrians, and Arup´s formula estimated accurately the critical number of pedestrians (73 pedestrians), it was not possible to evaluate the footbridge response in the postlock-in stage, as the Hivoss load did not include the auto-induced component of the pedestrian action. Moreover, the guide criterion for checking the lock-in based on the acceleration (0.1–0.15 m/s$^2$) gave a value of 167 pedestrians, which was far from the experimental result.

Similarly, Figure 11 shows the results of the full-scale crowd pedestrian tests that were performed on the Lardal footbridge, together with the numerical values obtained by applying the proposed method and by following the Hivoss guideline [12]. It can be observed that in the prelock-in stage the proposed method estimated the footbridge response less accurately than the Hivoss guideline, particularly for low pedestrian traffic (less than five pedestrians). However, it accurately predicted the lock-in point (around 13 pedestrians), while the two criteria of the Hivoss guideline gave poor results (31 and four pedestrians).

In addition, the proposed method predicted accurately the response in the postlock-in stage, as the analytical line was in good agreement with the experiments, while it was not possible to evaluate the footbridge response in this stage by using the Hivoss guideline.

Hence, it can be concluded that the proposed method provided a more complete and accurate procedure for estimating the lateral acceleration response of both footbridges than the Hivoss guideline, as it predicted successfully the lock-in point and the footbridge response with HSI. However, the Hivoss guideline gave a better approach in the prelock-in stage with light traffic.

## 4. Conclusions

This paper describes a simple but general formulation which permits the footbridge lateral response to be determined and thus the different stages of the lock-in phenomenon to be reproduced numerically. According to the results of this paper, the main contributions of this proposal are the following: (i) the method allows the lateral response of slender footbridges under postlock-in conditions to be estimated, which implies an important advantage over most current design guidelines; (ii) the method allows the footbridge comfort level to be evaluated once the lock-in phenomenon has developed and thus it permits assistance in the design of slender footbridges; and (iii) the method, based on a frequency domain approach, allows the lateral footbridge response to be computed in a simple way, whilst achieving a good agreement with the experimental data of the two footbridges analysed in the paper.

The performance of the method was successfully checked through the numerical response simulations of the Pedro e Inês footbridge and the Lardal footbridge, as the numerical results were in good agreement with the experimental full-scale data and the lock-in point was accurately calculated. However, it would be convenient to conduct full-scale crowd pedestrian tests on different typologies, to provide further confirmation of the method, and additional laboratory campaigns, to verify the current experimental expressions of the pedestrian loads.

**Author Contributions:** Conceptualisation, R.G.C. and J.F.J.-A.; methodology, R.G.C. and F.M.; validation, R.G.C., J.F.J.-A. and F.M.; formal analysis, R.G.C.; investigation, R.G.C.; resources, J.F.J.-A. and F.M.; data curation, R.G.C.; writing—original draft preparation, R.G.C.; writing—review and editing, R.G.C., F.M. and I.M.D.; visualisation, R.G.C.; supervision, I.M.D.; project administration, I.M.D.; funding acquisition, I.M.D. All authors have read and agreed to the published version of the manuscript.

**Funding:** The authors acknowledge the financial support provided by the Ministry of Science, Innovation and Universities (Government of Spain) by funding the Research Project SEED-SD (RTI2018-099639-B-I00).

**Conflicts of Interest:** The authors declare no conflict of interest.

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
