# Peer review of "Assessment of the Lateral Vibration Serviceability Limit State of Slender Footbridges Including the Postlock-in Behaviour"

_applsci, doi:10.3390/app10030967_

Round 1

Reviewer 1 Report

Τhe introduction provides sufficient background. Anyway, the issue in not new in its philosophy experience. There are examples of good practices all over the world.

Τhe introduction includes many relevant references, some of them in excess. It could be made some more recently published bibliography, not necessarily from Portuguese

The research is rather well designed to fulfil the goal of modelling the response of structures like footbridges under the weight of pedestrians.

The method followed is well-described. It permits evaluating the comfort level of the footbridge. Moreover, the lateral response two structures tested is calculated using the HIVOSS guideline and compared with the proposed method of the authors.

The results are analytically presented, helping the reader to have the whole picture of the testing techniques followed.

The conclusions are rather general, but many explanation views in the paper support the results, and vice versa.

There is not a need for better English, minor spell check required

Try to have a second look in references - There are mistakes in the citing of them (eg. 22)

Reviewer 2 Report

This paper describes a simple formulation that permits to determine the lateral footbridge response and thus to reproduce analytically the stages of the response. These stages have been observed in full-scale pedestrian tests performed in two structures.

The positive aspect of this paper can be found in obtaining a formulation that allows analyzing the different load scenarios that the footbridge will experience during its lifetime. Negative aspects are related to the lack of detailed explanation. Furthermore, the paper has a lack of comments and discussion about crucial aspects.

Comments:

There are some phrases that authors need to clarify. On page 2 line 71 :” ….similar to those applied on a rigid static surface”. What is rigid static surface? The authors need to clarify this expression. Page 2, line 72: “Lock-in: it is the instability point, in which the acceleration response builds up suddenly above the expected value.” What is “expected value”? Is there is any numerical value recommended for this phrase in any design code or guideline? If yes, the authors have to add a reference in the manuscript. Page 3 line 77: “The auto-induced component of the load must be included when evaluating the footbridge response to pedestrian action.” The authors need to clarify what is auto-induced component. Page 5, line 185, why the step frequency interval is considered between (μ-3σ, μ+3σ). What is the effect of this selection interval on the final results, if the step frequency interval considers more or less than this interval selection? In Figure 2, the authors have stated (0.4 £Fb £3 ). The authors need to explain about this interval. Moreover in page 6 line 221: “Figure 2 is defined from the dynamic tests performed by Ingólfsson et al. in 2009 at the University of Florence in Prato, Italy.” If the selected interval (0.4 £Fb £1.3 ) is derived from reference 31, it has to mention in figure 2 as well. After all equations, all coefficients have to be defined. After Eq. 2 what are cp, fp? In Eq. 8, what are Yn (fb) and H(fb). The authors need to explain how Eq. 6 converted to Eq, 7. This part of the manuscript is Vague and needs more explanation. Page 8, line 288, the authors need to clarify what is “W”. The table does not have a title. The authors need to justify why they considered j between 1-5. Page 9 line 306: “… in the case of a footbridge with fb=0.9 Hz, M=165,880 kg, K=5,304,000 Nm-1 , C=10,880 Nsm-1 , L=144 m and Ld=88 307 m.” These values belongs to a real case study. Page 10 line 326: “…pedestrians is like considering a “frozen moment” in which….”The authors need to clarify what is the “frozen moment”. Page 10 lines 332-344, the proposed formulation is not well explained. In the reviewer's opinion, it is better to explain the proposed formulation with a flowchart. In the opinion of the reviewer, only 2 case study is not enough to justify the proposed formulation.

Round 2

Reviewer 2 Report

I believe the manuscript has been significantly improved and it is ready for publication in Applied Sciences.